# Dietary Intervention with Cottonseed and Olive Oil Differentially Affect the Circulating Lipidome and Immunoregulatory Compounds—A Randomized Clinical Trial

**DOI:** 10.3390/metabo15090599

**Published:** 2025-09-08

**Authors:** Gwendolyn Cooper, Prabina Bhattarai, Brett Sather, Marguerite L. Bailey, Morgan Chamberlin, Mary Miles, Brian Bothner

**Affiliations:** 1Department of Chemistry and Biochemistry, Montana State University, Bozeman, MT 59717, USA; gwendolyncooper@montana.edu (G.C.); brettsather@montana.edu (B.S.); marguerite.bailey@student.montana.edu (M.L.B.); 2Department of Food Systems, Nutrition, and Kinesiology, Montana State University, Bozeman, MT 59717, USA; prabina.bhattarai@student.montana.edu (P.B.); mchamberlin1@billingsclinic.org (M.C.)

**Keywords:** metabolomics, lipidomics, cottonseed oil, dietary intervention

## Abstract

**Background/Objectives**: Cottonseed oil (CSO) is a dietary oil especially high in the n-6 polyunsaturated fatty acid (PUFA), linoleic acid (FA 18:2), which is a precursor for many pro-inflammatory eicosanoids. Curiously, diets rich in CSO have not been shown to cause increases in inflammatory markers or other negative health outcomes in humans. To rigorously test this, we have compared the health impact of a diet rich in CSO to olive oil (OO), which is generally considered to be a healthy oil. **Methods**: Specifically, this study examines circulating metabolite and lipid profiles during a 4-week dietary intervention with CSO or OO on 47 healthy adults. Untargeted metabolomics, targeted bulk lipidomics, and targeted lipid mediator analyses were conducted on fasting plasma samples taken pre- and post-dietary intervention. **Results**: A high degree of similarity was observed in the global metabolomic profiles of CSO and OO participants, indicating that CSO may elicit metabolic responses comparable to those of OO, potentially supporting similar effects on metabolic health markers. Targeted bulk lipidomics revealed changes in acyl chain composition reflective of the dominant fatty acid consumed—either 18:2 in CSO or 18:1 in OO. Immunoregulatory lipids 15-deoxy-PGJ2 and prostaglandin F2 alpha (PGF2a) were both higher in abundance in high-CSO diets, demonstrating differential effects of CSO and OO on immunoregulatory compounds. A correlative network analysis revealed two clusters arising from the dietary intervention as drivers of the dietary and immune responses. **Conclusions**: This study shows that CSO and OO differentially impact the circulating lipidome and immunoregulatory compounds in healthy adults.

## 1. Introduction

Cardiovascular disease (CVD) is a multifactorial condition that is the leading cause of death and chronic disability in the world [1,2]. Methods for reducing the risk of disease are heavily focused on diet and lifestyle modifications. Historically, the American Heart Association has encouraged reducing sugar and saturated fat intake, as well as increasing fiber, vegetable, and whole grain intake to reduce CVD risk [2]. The Mediterranean diet, which emphasizes plant-based foods and up to eight servings of olive oil daily, has also been recognized as a dietary intervention capable of reducing CVD [3,4]. The Mediterranean diet does not limit calories or cut entire food groups out, but instead emphasizes consuming an abundance of plant-based foods and fresh fruit daily, as well as olive oil as the primary fat source, limits dairy and red meat consumption, and suggests moderate amounts of fish and poultry [3]. The beneficial health effects of a Mediterranean diet are often contributed to olive oil (OO), which is high in the monounsaturated (MUFA) fat and oleic acid (FA 18:1) [4], in addition to other bioactive antioxidants and polyphenols, such as oleocanthal, oleuropein, and tocopherols [5,6,7]. Therefore, the Mediterranean diet, and thereby OO, is accepted as a healthy fat choice. Shifting people to a Mediterranean diet that includes replacing saturated fats with polyunsaturated fats has been shown to be one of the most effective ways to lower CVD risk [8].

Fatty acids are long-chain lipids with a variety of biological roles. Fatty acids act as the main structural components within membranes and can also be utilized for energy production and cellular signaling [9,10]. Fatty acid function is tied to their structure: fatty acids are classified as saturated (SFAs), monounsaturated or polyunsaturated (PUFAs), based on the number of double bonds present. Polyunsaturated fatty acids (PUFAs) are the chief structural components in cell membranes, however, they are also tightly connected to the immune system, where they are implicated in chronic illnesses, autoimmune reactions, and general health and wellness [11]. Among PUFAs, the omega-3 and omega-6 fatty acids derived from alpha-linoleic acid (ALA) and linoleic acid (LA), respectively, are essential fatty acids that must be consumed through diet since they cannot be synthesized endogenously [11]. In humans, these fatty acids give rise to arachidonic acid (AA), eicosapentaenoic acid (EPA), and docosahexaenoic acid (DHA), which can be further metabolized to bioactive signaling molecules called lipid mediators (LMs) [12]. Downstream products of the parent PUFAs regulate diverse homeostatic and immunomodulatory processes.

Diets high in n-6 PUFAs are traditionally associated with inflammation [13,14], platelet aggregation, vasoconstriction, and a wide range of diseases, including metabolic syndrome, chronic inflammation, and Alzheimer’s disease [13,14,15,16]. This association is attributed, at least in part, to the activity of pro-inflammatory lipid mediators made from n-6 PUFAs. Conversely, n-3 PUFAs are primarily involved in the resolution of inflammation [14,17]. In light of these pro-inflammatory vs. anti-inflammatory effects, lowering the ratio of n-6 to n-3 PUFA consumption to a 4–5:1 ratio is recommended to manage disease, including obesity and CVD [18,19,20].

Cottonseed oil (CSO) is a dietary oil especially high in the n-6 PUFA linoleic acid (18:2), for which the health impacts have not been adequately elucidated. Curiously, diets rich in CSO fed to rats resulted in an overall decrease in HDL cholesterol, despite an increase in saturated fats in the adipose tissue [21,22], whereas contrasting results have been reported from human studies. For example, normocholesterolemic adults who were administered a diet rich in CSO for 5–7 days saw an overall improvement in cholesterol levels [23,24]. In a longer study, administering a CSO-rich diet to participants with high cholesterol for 8 weeks resulted in significant improvements in both fasting and postprandial blood lipids [25]. This finding is especially interesting since linoleic acid is typically categorized as atherogenic due to its prooxidative and pro-inflammatory properties. Therefore, CSO presents a nutritional paradox. Due to CSO’s high linoleic acid content, high CSO consumption is expected to be associated with negative health outcomes, yet this does not appear to be the case.

To clarify the impact of CSO consumption, recent studies have re-examined the pro-inflammatory effects of n-6 PUFAs, particularly linoleic acid (LA). For example, mice fed high SFA, MUFA, and high-fat diets for 4 weeks had increased pro-inflammatory markers in the liver and adipose tissue; however, mice on a diet high in LA for four weeks did not have any significant changes in inflammatory or pro-coagulant markers in epididymal fat, liver, or plasma [26]. Thus, LA alone was not enough to induce an inflammatory response in the mice. Another study investigating associations between n-6 PUFAs in patient plasma and mortality in coronary heart disease in 2792 adults over 65 years old found that LA, but not other n-6 PUFA, was actually inversely associated with overall and coronary heart disease mortality [27].

The LA, and thereby CSO paradox is well documented in murine models and humans; however, the mechanism responsible remains a mystery. A particular gap in knowledge is the effect of CSO on the metabolomic and lipidomic profiles of humans. The biochemical actions of metabolites and lipids are far reaching and provide direct insight into cellular functions such as signal transduction, energy storage and production, immune regulation, and cellular health [28,29]. Nutritional epidemiological studies are beginning to utilize these -omics technologies in conjunction with traditional dietary assessment methods to deepen the knowledge of biochemical pathways impacted by nutrition, identify nutritional biomarkers and states of perturbation [30]. Though different types of instrumentation can be utilized for metabolomic and lipidomic analyses, liquid chromatography, coupled to tandem mass spectrometry (LC-MS/MS), is a high-throughput platform that can measure thousands of features with high resolving power, sensitivity, and specificity [31,32]. Not only can LC-MS/MS metabolomics identify significantly more compounds than traditional techniques such as the Seahorse Assay or NMR, but the inherent sensitivity of mass spectrometry allows for the measurement of low abundance compounds whose changes may be subtle, but whose impacts are profound [33,34].

The aim of this study was to compare the metabolic response of a four-week dietary intervention of high- and low-dose CSO versus OO on healthy adults via multi-omic analyses. We hypothesized that a 4-week intervention of CSO and OO will result in significant differences in the global metabolome and lipidome, as well as differences in metabolized products of arachidonic and linoleic acid. Untargeted and targeted metabolomic analyses were performed in tandem to provide a comprehensive representation of changes induced by the dietary interventions. To our understanding, this is the first study examining the effects of CSO and OO dietary interventions on the circulating lipidome and metabolome of healthy adults. This type of analysis demonstrates how CSO and OO dietary interventions impact the small molecule profile of healthy adults and how these data can be utilized to uncover the complexities of food intake on human physiology.

## 2. Materials and Methods

### 2.1. Study Population

Study participants were recruited in Bozeman, Montana (MT), from August 2022 to November 2023. Inclusion criteria required participants to be between the ages of 18 and 55 years old, with a body mass index of 18–27 kg/m^2^. Exclusion criteria included the following: medications that lowered cholesterol, lipids, and/or inflammation, blood pressure medications, antioxidant supplementation (vitamin E or C or iron) over 800 IU/day, oral contraceptive use, diabetes or gallbladder conditions, smoking within the last 30 days, consumption of alcohol exceeding 14 drinks/week, allergies to dairy, egg, tree nuts, peanuts, shellfish, soy, fish, wheat, legumes, olive oil, or cottonseed oil, ketogenic or paleo diet within the past six weeks, pregnant or lactating women, those on a weight-loss diet or who had plans to change their exercise regimen, those with physical mobility issues, or any other health concerns or conditions that could interfere with this study. the screening of potential participants was conducted using REDCap (version 13.10.6) [35], American Heart Association/American College of Sports Medicine (AHA/ACSM) Health/Fitness Facility Preparation Screening Questionnaire [36], and the Physical Activity Readiness for Everyone Questionnaire (PAR-Q+) [37]. In total, 60 subjects met the eligibility requirements and were invited to participate in this study. Of the 60 participants invited, 47 individuals completed this study and were included in all subsequent analyses.

### 2.2. Experimental Design

This study was a randomized, double-blind, parallel clinical trial with four conditions: high-dose cottonseed oil, low-dose cottonseed oil, high-dose olive oil, and low-dose olive oil. The USDA Dietary Guideline for Americans recommends 27 g of oil per 2000 kcal for healthy adults [38]. Therefore, our dosing regimen consisted of 30 g of oil for the low-dose groups and 60 g of oil for the high-dose groups. The study protocol was approved by the Institutional Review Board at Montana State University (2022-146; approved on 29 July 2022). This study was prospectively registered with ClinicalTrial.gov (NCT05439590) on 27 June 2022.

Participants came to the Nutrition Research Laboratory on three occasions for testing. During the first visit, informed consent was reviewed and obtained prior to completing general health questionnaires and anthropometric measurements. At visit two, anthropometric measurements and fasting blood samples were collected. At the end of the second visit, participants were randomized to one of the four diet arms and provided accompanying diet guidelines to follow during the 4-week intervention period. Participants then returned to the lab for repeated testing during the third visit. See Appendix A for a summary of the study design and Appendix A for CONSORT diagram. Plasma samples taken before and after the dietary intervention were analyzed via LC-MS-based untargeted metabolomics and targeted lipidomics to evaluate differences in the metabolomic profiles of the four groups.

The R package, Gpower (version 3.1), was utilized to calculate the power a posteriori. A total sample size of 47–48 participants was determined to detect a medium effect size using repeated-measures ANOVA with 79–80% power for detecting changes in total cholesterol levels.

### 2.3. Dietary Intervention and Adherence

A randomized protocol assigned participants to one of the four experimental groups. Block randomization was performed using the blockrand function in R (version 1.5). Researchers, volunteers, and clinical staff participating in data collection were blinded to condition assignments at visit 2. Allocation sequence was determined by a third party.

The participants were asked to consume two doses of smoothies—one mango-flavored and one chocolate-flavored every day during the 4-week intervention period (28–30 days) containing 30 g or 60 g of CSO or OO based on group assignment. Participants were asked to consume two smaller smoothies a day rather than one large smoothie in order to avoid issues with consuming too much in one sitting, i.e., 800 kcal and 60 g of fat in one sitting do not appropriately mimic fat intake across multiple meals a day. A registered dietician designed and prepared the study smoothies. Smoothie flavors within each dosing regimen were matched for protein, carbohydrate, fat, and calorie content. For example, both the 30 g CSO and the 30 g OO chocolate-flavored smoothies contained 3.81 g of protein. While the protein content of mango and chocolate smoothies differed by a small amount, participants had one of each flavor daily to ensure that the protein from smoothies was the same within each oil dose per day. Two flavors were included in this study to provide some variety and increase compliance. Detailed information on the nutritional composition of the smoothies can be found in Appendix A. The smoothies were prepped in plastic to-go containers and frozen for pick-up. The participants were instructed to eat two smoothies a day, but no specific time was required for ingestion. The participants were instructed to avoid any fried foods, salad dressings containing oils/oilseeds, hummus, nutrition bars containing oils/oilseeds, peanut butter, or other foods containing nuts/oils/oilseeds, and omega-3 fatty acid supplements and other dietary supplements (fish oil, vitamin E, and vitamin C) throughout this study period. Additionally, the participants were instructed to limit nuts and oil seeds to one serving (42.5 g) per week, restrict fatty meats, sausages, and fatty fish to one serving per week, and limit cheese, full-fat milk, egg yolks, and butter throughout the study period. The participants were also given a bottle of either CSO or OO depending on their grouping to cook with throughout the study period.

Adherence to smoothie consumption was monitored during smoothie pick-ups, which occurred once per week or every two weeks, depending on participant preference. The researchers recorded the type and number of smoothies consumed by participants every week. Participants were excluded from the analysis if adherence failures exceeded one day.

### 2.4. Blood Sample Collection

Prior to visits two and three, the participants fasted and avoided strenuous activity for 12 h and alcohol consumption for 24 h. A standard venipuncture technique, performed by a phlebotomist or medical doctor, from an antebrachial vein was used for blood collection. Briefly, the skin was cleaned using an alcohol swab, and blood was collected through a single-use needle into vacutainer tubes. Blood markers, including total cholesterol (TC), high-density lipoprotein cholesterol (HDL), low-density lipoprotein cholesterol (LDL), glucose (GLU), and triglycerides (TG), were measured using whole blood collected in heparin tubes on a Picollo Xpress Chemistry Analyzer lipid panel (Abaxis, Union City, CA, USA). Plasma was frozen at –80 °C until subsequent mass spectrometry and cytokine analyses, respectively. An Ella Automated Immunoassay system (BioTechne, USA) was utilized to measure interleukin 1-beta (IL-1β), interleukin-6 (IL-6), interleukin-8 (IL-8), interleukin-10 (IL-10), tumor necrosis factor alpha (TNF-alpha), interferon-gamma (IFNg), and interleukin-1 receptor alpha (IL-1Ra). Segmental multifrequency bioelectrical impendence analysis (Seca mBCA 515, Hamburg, Germany) was utilized to assess anthropometrics. General linear models, including ANOVA and two-tailed t-tests, were used to assess any potential differences in the basic lipid panel and inflammatory markers between CSO and OO groups.

### 2.5. Fatty Acid Analysis of Oil Samples via Mass Spectrometry

To determine the fatty acid composition of the oils, a fatty acid methyl esterification (FAME) protocol was followed. Concentrated HCl was diluted with methanol (35%, w/w) to make methanolic HCl. This was diluted to a final concentration of 85% methanol and 15% water (v/v). An amount of 2 mL of the methanolic HCl was added to 100 µL of each oil sample. The samples were then vortexed for 30 s and placed in an incubator set to 37 °C overnight to ensure complete esterification of the fatty acids. After cooling to room temperature, 1 mL of hexanes and 1 mL of water was added. The samples were vortexed briefly and left for 5 min to allow for phase separation to occur. The top layer containing the FAMEs was extracted and then dried down using N_2_. Once dried, the samples were resuspended in hexane for subsequent gas chromatography mass spectrometry (GC-MS) analysis.

An Agilent 7890A/5975C XL GC-MS system, equipped with an HP-5MS 5% phenyl methyl siloxane column (Agilent) and an Agilent 7693 autosampler, was utilized for analysis of FAMEs extracted from the oils. The 20 min method began with an oven temperature of 45 °C and was increased by 20 °C/min until a final temperature of 325 °C was reached. The obtained data were analyzed using Agilent Mass Hunter (10.0). National Institute of Standards and Technology (NIST) library version 11 and an in-house library were utilized and cross-referenced for the identification of mass spectra collected. Peak integration and the subsequent area under the curve (AUC) were calculated using a combination of Agilent Mass hunter software (version 12.1) and manual peak verifications.

### 2.6. Untargeted Metabolomic Analysis of Plasma

To extract plasma metabolites and precipitate proteins, 100 µL of plasma was added to 400 µL of cold acetone. The samples were then placed at −80 °C overnight. To isolate the metabolites from proteins and other macromolecules, the samples were centrifuged at 16,100× *g* for 15 min at 4 °C. Supernatants consisting of metabolites were then collected and dried via vacuum concentration. Prior to mass spectrometry analysis, the dried samples were resuspended in 100 µL of 1:1 acetonitrile/water. At this time, pooled samples, consisting of 2 µL of resuspended metabolites from each sample, were made.

Following metabolite extraction, all samples were analyzed by liquid chromatography–mass spectrometry (LC-MS) using a Waters Acquity UPLC coupled through an electrospray ionization source to a Waters Synapt XS. A Cogent Diamond Hydride HILIC column (150 × 2.1 mm) at a flow rate of 400 µL/min was used to separate the metabolites. Solvent A was 95% water 5% acetonitrile with 0.1% formic acid, and solvent B was 95% acetonitrile 5% water with 0.1% formic acid. The 19 min elution gradient decreased from 95% to 25% solvent B over 12 min, followed by a 5 min wash of 25% solvent B. Each run began with 2 min of washing. Blanks and pooled samples were injected every 10 samples throughout the run to track spectral drift and to assess LCMS performance. All samples underwent standard MS1 and the pooled samples underwent liquid chromatography–tandem mass spectrometry (LC-MS/MS) with a constant energy ramp of 20–50 V. MS^E^ data were collected for pooled samples to assist in metabolite identification. All data were inspected manually to determine if any issues arose during the run.

Data processing, including peak picking and alignment, was conducted utilizing Water’s Progenesis QI software, v.2.3 (Nonlinear Dynamics, Newcastle, UK). Progenesis and an in-house metabolite library (Mass Spectrometry Library of Standards, IORA Technologies, Ann Arbor, MI, USA) were utilized for metabolite annotations. The Human Metabolome Database, ChemSpider, and the in-house metabolite library were used for comparing acquired and theoretical fragmentation of MS/MS data. Multivariate statistical analysis was performed using MetaboAnalyst 6.0 https://www.metaboanalyst.ca/home.xhtml (accessed on 1 January 2025). Standard procedures [39] were applied for correcting non-normal distributions, while raw data were log-transformed, quantile-normalized, and auto-scaled prior to analysis.

### 2.7. Targeted Lipid Mediator (LM) and Total Bulk Lipidomic (TBL) Analyses

For all LCMS methods LCMS grade solvents were used. An amount of 250 µL aliquots of each plasma sample were immersed in 250 µL of ice-cold methanol. Following this, 250 µL of water and 400 µL of chloroform were added. Samples were shaken for 20 min at 4 °C, then centrifuged at 16,000× *g* for 20 min. An amount of 250 µL each of the top (aqueous) and bottom (organic) layers were collected separately. The aqueous layer was diluted 5× in 50% methanol in water for LCMS analysis of oxidized lipid mediators. The organic layer was dried down under vacuum and resuspended in an equivalent volume of 5 µg/mL butylated hydroxytoluene in 6:1 isopropanol/methanol for targeted bulk lipidomics. Tributylamine and all molecular references compounds were purchased from Millipore Sigma. LCMS grade water, methanol, isopropanol, and acetic acid were purchased through Fisher Scientific.

Oxidized lipid mediators (LMs) were analyzed using the same MRMs as previously described but without solid phase extraction [40]. The samples were analyzed using a LD40 X3 UHPLC (Shimadzu Co., Columbia, MD, USA) and a 7500 QTrap mass spectrometer (AB Sciex Pte. Ltd., Framingham, MA, USA). An aliquot of the aqueous layer of the initial metabolite extraction was separated on a Phenomenex Kinetex Polar C18 (100 Å, 2.6 µm, 3 × 100 mm) using a binary gradient of (A) 0.01% acetic acid in water and (B) 0.01% acetic acid in methanol over 11.5 min. Samples were detected in negative mode with triggered enhanced-product ion scans for post-hoc spectral identification. Spectral information was compared with standards and a spectral library for identification. A blank and a standard mix were serially injected every 10 injections. The standard mix consisted of RvE1, LXA4, LXA5, LXB4, PGE2, PGD2, PGF2a, PGJ2, TxB2, PD1, PDX, RvD5, Maresin 1, LTB4, 5,15-DiHETE, 14-HDHA, 18-HEPE, 13-HODE, 12(13)-EpOME, 9(10)-EpOME, 12,13-DiHOME, 9,10-DiHOME, arachidonic acid (AA), EPA, DPA, and DHA (each at 10 ng/mL).

Bulk lipidomics was performed as previously described [41] with a shortened gradient and polarity switching. A LD40 X3 UHPLC (Shimadzu Co.) and a 7500 QTrap mass spectrometer (AB Sciex Pte. Ltd.) were used for separation and detection. Lipids were chromatographically separated on a Waters XBridge Amide column (3.5 μm, 3 mm × 100 mm) using a 9 min binary gradient, starting from 100% 5 mM ammonium acetate with 5% water in acetonitrile (apparent pH 8.4) and ending at 95% 5 mM ammonium acetate, 50% water in acetonitrile (apparent pH 8.0). All lipids were detected using scheduled MRMs that leveraged fatty acid product ions, headgroup product ions, or neutral loss ion combinations that are conserved within each lipid class.

All lipidomic and LM signals were processed and integrated using SciexOS 3.1 (AB Sciex Pte. Ltd.). Signals with greater than 50% missing values for a specific tissue set were discarded and the remaining missing values were replaced with the lowest registered signal value. Signals with a QC coefficient of variance greater than 30% were discarded. Lipids and LMs with multiple MRMs were quantified with the higher signal-to-noise MRM. Filtered bulk lipidomics data were total-sum normalized after initial filtering. Multivariate statistical analysis was performed using MetaboAnalyst 6.0 https://www.metaboanalyst.ca/home.xhtml (accessed on 1 January 2025) and R (version 4.4.1).

### 2.8. Network Analysis

To determine whether relationships existed among the lipids, LMs, and clinical covariates, a correlation-based network analysis was conducted. Correlation-based network analysis utilizes mathematically defined correlations to reflect the magnitude of similarity between variables in datasets [42]. These relationships are then visualized via a network analysis to assist in the biological interpretation of complex data. To reduce noise and pin-point relationships between significant omics and clinical data, only significant (*p* < 0.05) or trending (*p* < 0.1) TAGs and LMs from the respective analyses were utilized. The in-house-developed R (version 4.4.1) code for correlation matrix calculation and network generation are publicly available on GitHub https://github.com/IDSS-NIAID/MetLipNet (accessed on 6 June 2025). The “rcorr” function from the Hmisc package was leveraged to calculate correlation matrices from integrated datasets consisting of metabolomics, lipidomics, and clinical variables. The “cor_gather” function from the rstatix package facilitated the coercion of correlation matrices to a format that was compatible with the igraph package for network visualization. Specific nodes of interest were interrogated by examining their first- and second-degree neighbors using the “ego” function from the igraph package [43]. Networks were plotted using the ggraph package.

## 3. Results

### 3.1. Participant Characteristics

Between August 2022 and November 2023, a total of 163 adults were assessed for eligibility by completing a RedCAP survey, the AHA/ACSM Health/Fitness Facility Preparation Screening Questionnaire, and the PAR-Q+ questionnaire. Of the 163, 60 met the screening criteria and were invited to participate in this study. Of the 60 participants that enrolled, 13 withdrew from this study due to illness, lack of time, inability to drink smoothies, inability to follow dietary guidelines, or adverse events (Appendix A). In total, there were 47 participants that completed this study (24 male and 23 female). The average participant age was 31.49 ± 11.83 years, with an average BMI of 23.13 ± 1.98 kg/m^2^. Waist circumference (cm) and visceral adiposity (L) were significantly greater in males compared to females (*p* = 0.003, *p* = 0.003). Fasting glucose (mmol/L) and triglycerides (mmol/L) were also significantly greater in males compared to females (*p* = 0.02, *p* = 0.001 respectively) (Table 1).

### 3.2. Impact of CSO and OO on Global Metabolic Status

To investigate metabolic changes induced by dietary interventions of high- and low-dose CSO and OO, an untargeted metabolomic analysis was conducted on the plasma samples from the participants. To analyze global changes in the entire small-molecule profile, an acetone crash was utilized to isolate small molecules from proteins. Metabolites were separated via HILIC chromatography prior to being analyzed in positive mode via mass spectrometry. In total, 2675 metabolomic features were detected in the experiment. A paired analysis, consisting of a delta change of metabolite intensity (intensity measured in pre dietary intervention subtracted from post intervention), was performed for all participants to determine the change induced by the diet. To better visualize the potential differences between the groups, principal component analysis (PCA) was conducted between the low-dose and high-dose oil groups. There was virtually no separation between any of the groups indicating that there was not a global change in the metabolomic profiles due to the dietary intervention (Figure 1A,B).

Using a more targeted analysis of variance (ANOVA), 30 features changed significantly in abundance (*p* < 0.05) between all four dietary groups. A heatmap of the group averages for each of these metabolomic features was generated to determine if there were trends in features that changed with either the type of oil consumed (CSO vs. OO) or the dose (high vs. low) (Figure 1C). Interestingly, there were not large global changes induced by the two oils, as evidenced by the high degree of similarity in feature regulation patterns across all four groups. These data suggest there are very few differences between CSO and OO groups. Of the features that were identified, several were lipids and compounds involved in fatty acid transport, including dodecanoyl carnitine, diacylglycerol (18:1/16:0), and diacylglycerol (16:0/18:2). Given that the markers identified were lipid related, a more targeted lipidomic analysis was conducted to seek greater resolution in distinguishing metabolic differences between the experimental groups.

### 3.3. Fatty Acid Composition of CSO and OO

To determine the total fatty acid composition of the two dietary fats, a fatty acid methyl esterification (FAME) analysis on a GC-MS was performed. The dominant fatty acid in cottonseed oil was linoleic acid (FA 18:2), comprising 60% of the sample, followed by palmitic acid (FA 16:0) (28%), oleic acid (FA 18:1) (5%), stearic acid (FA 18:0) (2%), and palmitoleic acid (FA 16:1) (2%) (Figure 2A).

Olive oil was predominantly composed of oleic acid, comprising 51% of the sample, followed by palmitic acid (26%), stearic acid (12%), squalene (8%), and palmitoleic acid (2%). These results agree with reported values [44,45] and demonstrate that the oils themselves vary significantly in composition. Additional fatty acids, representing less than 1% of total fatty acids, were also detected (Appendix A).

### 3.4. Impact of CSO and OO on Lipidomic Profiles

Untargeted metabolomic analysis identified minor global differences in participant plasma samples, with relatively few features changing significantly (Figure 1). Of the features that did change, several were identified as lipids or were related to lipid metabolism (Figure 1C). Based on the differences in the fatty acid composition of the oils and the fact that most of the identified metabolomic features in the heatmap were lipid species, a targeted lipidomic analysis was performed on participant plasma. To isolate lipids from other metabolites, a Bligh–Dyer extraction was performed, and the organic layer was utilized for subsequent analysis. Lipids were separated on a Waters XBridge Amide column and analyzed on a QTrap mass spectrometer in positive and negative modes.

After data filtering, 794 unique lipids were detected. Since the magnitude of a diet-induced change was expected to be larger in the high-dose CSO versus the high-dose OO groups, a pairwise t-test was performed to evaluate which lipids changed in response to the high oil intervention. A total of 50 lipids showed modest change between high-dose CSO and the high-dose OO (*p* < 0.1). To evaluate the differences between these lipids across all groups, they were plotted via a heatmap (Appendix A). Strikingly, the normalized abundance of the top lipids in the high-dose CSO and high-dose OO groups were almost always opposite. Lysosphingolipids, ceramides, and phosphatidylinositols decreased in abundance in the high-dose CSO group and, to a lesser extent, in the low-dose CSO group compared to the decreasing abundances in the OO groups. Curiously, triacylglycerides, phosphatidylcholines, and phosphatidylethanolamines had mixed abundance patterns in the high-dose CSO group, with some increasing and some decreasing in abundance.

The acyl chain composition of phospholipid classes has biological significance, reflecting processes including lipid remodeling, modulation of membrane trafficking, and selective hydrolysis [46,47]. To investigate changes in acyl chain composition, lipids were grouped according to their acyl chain composition at position 2 and visualized via a heatmap (Figure 2B). The participants that consumed the CSO diets had an enrichment in 18:2 acyl chain composition, in agreement with linoleic acid (FA 18:2) being the predominant fatty acid in cottonseed (Figure 2). Participants that consumed OO diets had enriched 18:1 acyl chain composition, reflecting the predominant fatty acid, oleic acid (FA 18:1), present in OO.

### 3.5. Immunoregulatory Compounds Affected by Dietary CSO and OO Interventions

To further probe changes in lipid profiles, specifically in bioactive lipids implicated in inflammatory and immunoregulatory processes, a targeted lipid mediator (LM) analysis was performed. In total, 65 LMs were detected in participant plasma (Appendix A). To identify differences between the groups, a PCA was conducted between the high-dose and the low-dose oil groups. Interestingly, there was a large degree of overlap between both the high-dose and the low-dose oil groups, indicating that the LM responses to CSO and OO were similar (Figure 3A,B).

To parse out any potential differences in LM profiles induced by the dietary intervention, an ANOVA was performed. Two LMs changed significantly, 15-deoxy-PGJ2 and PGF2a, both of which increased in abundance in the CSO groups compared to the OO groups (Figure 3C). This was of particular interest, given that both 15-deoxy-PGJ2 and PGF2a are products of the arachidonic acid (AA) pathway, with very different biological actions [48,49].

### 3.6. Correlations Between Oil Consumption, Clinical Measurements, and -Omics Data

To determine if relationships existed between the general lipid pool, LMs, and the clinical covariates measured, a network analysis was performed. The relationships between covariates were formalized as a network where lipids, LMs, and clinical measurements corresponded to nodes. Nodes are connected by edges if a relationship exists between them. Correlation analysis is a powerful tool for identifying relationships in complex systems that may be missed by approaches such as ANOVA and hierarchical clustering [50]. This technique was used to examine relationships between the high-oil groups, as the largest changes or highest likelihood of differences were anticipated in these groups.

The network generated contained three clusters (Appendix A), consisting of 48 nodes—all of which have a correlation coefficient greater than 0.5 and a *p*-value less than 0.01. As we were interested primarily in the relationships between the omics datasets and clinical covariates, clusters 1 and 2 were prioritized. The final network (Figure 4) consisted of two clusters and represented LMs, lipids, and clinical measurements.

Cluster 1 is dominated by positive correlations of triglycerides (TAGs), many of which have a linoleic acid acyl chain (Figure 4B). This confirms a strong impact of the dietary intervention on circulating TAGs. Notably, the TAG species exhibit negative correlations with body mass index (BMI), clinical triglyceride levels, very low-density lipoprotein (VLDL) concentrations, average waist circumference, and average body weight, while showing positive correlations with total cholesterol and low-density lipoprotein (LDL) levels. The positive correlation between the TAG species and total cholesterol was anticipated, as it reflects established relationships between anthropometric values and blood lipids [51,52]; however, the negative correlation between the TAG species and clinical triglyceride levels was surprising, as these are lipids belonging to the same functional lipid class. Additionally, the only LM, PGF2a, was found to negatively correlate with average fat mass and average body weight. These relationships between dietary lipids (TAGs), bioactive lipids (LMs), and clinical measurements demonstrate the ability of cottonseed oil and olive oil to affect the circulating lipidome and immunoregulatory compounds. Cluster 2 (Figure 4C) is primarily composed of TAGs, with oleic and stearic acid acyl chains, which are positively correlated with each other. Curiously, the other significantly regulated LM, 15-deoxy-PGJ2, clusters with and is negatively correlated with this set of TAGs. The cytokine IL-1RA is also negatively correlated with the TAGs within this cluster.

## 4. Discussion

The results of this study reveal changes in the circulating lipidome and immunoregulatory compounds induced by a 4-week dietary intervention with CSO or OO. Interestingly, the global metabolome of the CSO participants was very similar to that of the OO participants, indicating that overall response is similar for both types of oil. One specific change observed was that plasma lipid acyl chain composition reflected 18:2, linoleic acid, if CSO was consumed, or 18:1, oleic acid, if OO was consumed. This is a clear indication that the fats consumed directly influence the pool of lipids in circulation. Network analysis revealed two clusters of variables that responded to the CSO and OO interventions: one descriptive of the bulk drivers of diet and one descriptive of the bulk drivers of the immune response. Together, these findings highlight the complexity of food systems and their impact on human metabolic status.

### 4.1. Metabolomic Profiles of CSO and OO Are Similar

We found that the metabolomes of participants on low-dose and high-dose CSO and OO were very similar (Figure 1). The limited observed differences between these two diets are not wholly unexpected, as all study participants were healthy at baseline (Table 1 and Appendix A), and this result reflects the much greater impact of overall diet versus the relatively small changes in oil consumption implemented in this intervention.

Other studies observing gross changes resulting from CSO interventions were conducted with relatively short intervention periods (5–7 days) [24,53] or consisted of patients with health perturbations such as dyslipidemia at baseline [54]. As research into n-6 PUFAs continues, it is becoming more evident that a “one size fits all” model of the alleged pro-inflammatory and negative consequences of n-6 PUFAs such as linoleic acid may not be appropriate [55]. Further, these results suggest that CSO could be re-examined as a potential healthy alternative fat source, given the similarities in metabolic responses between CSO and OO.

This study is the first of its kind to utilize an ultra-sensitive, global profiling technique to characterize human responses to 4-week dietary interventions with CSO and OO. By employing an untargeted metabolomics approach, we were able to determine that the metabolic response of the OO and CSO participants was relatively similar, with only minor differences being observed, primarily in lipid and lipid-like molecules (Figure 1C). To further characterize these differences in lipids, a targeted lipidomic analysis was performed.

### 4.2. Decreases in Clinically Relevant Lipids and Acyl Chain Composition Are Observed in CSO Participants

After identifying subtle and interesting differences in the metabolome of the CSO and OO participants, we conducted a targeted bulk lipidomic analysis to determine if the changes were specific to the lipidome. Lysophosphatidylcholine (LPC) is a major phospholipid implicated in the induction and progression of atherosclerosis, is positively associated with cardiovascular and neurodegenerative diseases, and is an etiological factor in autoimmune diseases such as systemic lupus erythematosus [56,57,58]. Structurally similar to LPC, lysophosphatidylethanolamine (LPE) has been implicated in impaired lipid droplet catabolism and has the potential to induce fatty liver formation [59]. Additionally, ceramide (CER) levels are predictors of mortality in patients with stable coronary artery disease and acute coronary syndromes [60]. Therefore, decreases in LPC, LPE, and CER abundance are often associated with positive clinical outcomes. In this study, we observed a decrease in LPC, LPE, and CER abundance in the CSO participants when compared to the OO participants (Appendix A), suggesting that CSO has the potential to improve several clinically relevant lipid species. Though these results are promising, further research is necessary to confirm these findings.

To further investigate lipidome-specific changes induced by CSO and OO interventions, the top 30 lipids identified via the ANOVA analysis (Appendix A) were grouped according to the acyl chain composition rather than by functional group. Acyl-chain diversity is notably diverse and can be influenced by nutritional state, metabolic dysregulation, and genetic alterations [61]. Therefore, characterizing the acyl chain composition is a valuable tool for linking nutritional epidemiology to biological functions. This alternative grouping strategy revealed that the acyl chains at position 2 directly reflected the dominant fatty acid, as determined by the FAMES analysis (Figure 2). Therefore, these data demonstrate how the type of fat consumed is directly reflected in circulation and suggests that tracking the acyl chain composition of fats in circulation may be a method for assessing and tracking biomarkers of food intake.

The rapid effects on the circulating lipidome resulting from dietary oil supplementation demonstrated within this study also has implications for precision medicine. Several studies have emphasized the importance of tailoring personalized diets to acute metabolic responses to food intake [62,63]. We demonstrate that a four-week dietary intervention of CSO and OO induces global shifts in the circulating lipidome, which has the potential to be leveraged by physicians for alterations of real-world dietary behavior and dietary risk prevention. Additional studies with larger cohorts would be ideal for testing this hypothesis.

### 4.3. Immunoregulatory Lipids Are Differentially Affected by CSO and OO

Given that CSO and OO lipidomes were distinguishable, and that the dominant PUFAs in the oils were either the pro-inflammatory omega-6 or the anti-inflammatory omega-3, we performed a targeted lipid mediator analysis on participant plasma to see if there were differences in these immunoregulatory lipid species. Surprisingly, there was little spontaneous variation observed in the PCAs (Figure 3A,B), indicative of few differences between the groups. Although other studies have observed differences in PUFAs and their LM metabolites as a result of dietary intervention, these studies have primarily focused on omega-3 PUFAs [64,65] and in participants with pre-existing metabolic syndromes [64,66]. All participants in this study were healthy adults, which could explain the large degree of similarity between LM profiles.

The similarities in LM profiles between the n-6 rich CSO diet and the n-3 rich OO diet also demonstrates that a diet high in n-6 fatty acids is not inherently unhealthy. Traditionally, n-6 rich diets were thought to be unfavorable, since LA can be elongated to AA, which plays key roles in carcinogenesis, inflammatory diseases, and cardiovascular biology [67]. Importantly, several larger and more recent studies are changing how we perceive the effects of LA and n-6 fatty acids on human biology. In several large cohorts, no significant associations have been found between n-6 PUFAs and coronary disease, or inflammatory markers including c-reactive protein, fibrinogen, or tumor necrosis factor-a [68,69]. Our results agree with these findings, as we did not find that adding LA to the diet via CSO impacted AA or downstream pro-inflammatory eicosanoids.

Only two LMs, 15-deoxy-PGJ2 and PGF2a, changed significantly based on the one-way ANOVA analysis (Figure 3C), both of which increased in abundance in the high-CSO group. In mice fed high fat diets, PGF2a was responsible for an improvement in insulin sensitivity and signaling in peripheral cells [70]. Additionally, exogeneous administration of PGF2a decreased high-fat-diet-induced hepatic steatosis and immune cell infiltration, demonstrating that this prostaglandin has potent effects on body metabolism [70]. These findings are also applicable to humans. In three independent cohorts, low levels of PGF2a were associated with overweight and obese individuals [70]. The higher abundance of PGF2a in high-CSO participants could therefore indicate a positive impact of CSO on systemic metabolism and sensitivity. These findings are especially interesting, as PGF2a is widely accepted as eliciting pro-inflammatory responses [71], in addition to conflicting evidence on prostaglandins and insulin resistance [72,73].

The effects of 15-deoxy-PGJ2 are less divisive, and this LM is widely reported to exert anti-inflammatory effects on numerous tissue and cell types [74,75,76]. In one study examining cardiomyocytes, 15-deoxy-PGJ2 was shown to modify transcription of pro-inflammatory genes, including the downregulation of IL-6 production and upregulation of IL-8 production by suppression of NF-kB and MAPK pathways [76], suggestive of cardio protection by the LM. Within our study, the increased abundance of 15-deoxy-PGJ2 in response to high CSO consumption (Figure 3C) could indicate positive implications for CVD development and progression; however, given the dichotomy of the effects of LMs on human metabolism and health, additional research is warranted to draw more concrete conclusions. Regardless, it is evident that 4-week dietary interventions with CSO and OO differentially affected immunomodulatory lipids in healthy adults.

### 4.4. Bulk Drivers of Dietary Response and Immune Response Are Observed in Response to Interventions

A network analysis was conducted to continue the investigation of how immunomodulatory lipids, circulating lipids, and clinical variables related to the dietary intervention. Network analysis provides a useful way to compare how components of a system interact or relate to one another. These interactions are driven by statistical analysis, to highlight associations between components [77]. In this study, circulating lipids (Figure 2B) and LMs (Figure 3C) that were found to be significant were combined with clinical measurements into one complete dataset and analyzed via Spearman correlation analysis. The subsequent analysis was visualized via network analysis (Appendix A) which identified three dominant clusters of correlated variables. To improve data integration, leverage all data collected, and understand the relationships between clinical variables and the -omics measurements, only the clusters with both clinical and -omic measurements were further analyzed (Figure 4).

The network analysis highlighted two clusters of interest, indicating two different responses of the participants during the dietary intervention. Cluster 1 (Figure 4B) consists of numerous TAGs that are positively correlated with one another. Interestingly, these TAGs are negatively correlated with BMI, clinical triglycerides, VLDL, average waist circumference, and average body weight. The negative correlation between clinical triglycerides and lipidomics-measured TAGs could indicate that these TAGs may be functionally active components of the lipidome and not simply reflective of diet. TAGs provide the body with a rich and efficient energy source, requiring transport, storage, and repackaging, and their use is highly regulated [78]. Their dysregulation is tied to numerous clinical phenotypes, such as CVD and pancreatitis [78]. Though this link between diet and TAGs is well understood, recent advances have expanded their functionality through connections to seizures [79], cognition [80], and even the long-term COVID-19 syndrome [81]. The negative correlation between lipidomic and clinical TAGs indicates that although the participants were administered a TAG rich diet, this did not result in a bulk increase in TAGs. Overall, cluster 1 demonstrates that the effect on the lipidome induced by dietary CSO and OO interventions is divorced from the TAG response observed.

Network cluster 2 (Figure 4C) is also composed primarily of TAGs that are positively correlated to one another. Interestingly, there is a negative correlation between TAGs in cluster 2 and IL-1RA. IL-1RA is a member of the IL-1 family of cytokines and is an important anti-inflammatory protein in numerous diseases, such as cancer, osteoporosis, diabetes, and more [82,83]. In mice, IL-1RA has been shown to protect against high-fat-diet-induced insulin resistance and reduced local adipose tissue inflammation, despite immune cell recruitment [84]. Additionally, IL-1RA has been suggested as an early biomarker for progression to type 2 diabetes due to its close association with triglycerides [85,86]. Increasing IL-1RA levels are correlated with adiposity markers, including blood triglyceride levels, and has been shown to be a predictor of type 2 diabetes [85]. 5-deoxy-PGJ2 also exhibits a negative correlation with this set of TAGs. In rats, 15-deoxy-PGJ2 has been shown to affect peroxisome proliferator-activated receptors (PPARs), which are directly involved in fatty acid and lipid metabolism. Therefore, cluster 2 may be more indicative of the bulk drivers of the immune response, whereas cluster 1 is more reflective of the bulk drivers of the dietary response to the CSO and OO dietary interventions. Further research is necessary to understand the complex relationship between dietary interventions, PUFA metabolism, and small molecule composition.

### 4.5. Limitations

We acknowledge that there are additional factors and limitations of this study that must be addressed. Only healthy adults were allowed to participate, and therefore, the results discussed may not be applicable to individuals with increased metabolic risks or pre-existing metabolic perturbations (such as diabetes, hypercholesterolemia, etc.). In addition, the dietary instructions were quite strict and could have resulted in alterations in the daily eating habits of the participants. It is possible that alterations in the eating patterns of the participants could have impacted this study’s outcome measures. However, daily eating patterns were tracked via questionnaire both pre- and post-dietary intervention (Appendix A) and did not appear to result in significant shifts—indicating a relatively stable diet intake. The participants were also given additional CSO or OO to cook with. While the additional amount of oil consumed during cooking was not used to define or adjust participants’ assigned doses, it was tracked to ensure that the participants were not consuming far more than the amount assigned to their high- or low-dosage group. For the majority of participants, the amount of additional oil consumed was minimal (Appendix A) and therefore was unlikely to have a significant impact on the overall amount of CSO or OO consumed or on the measured results. An additional limitation of this study is its sample size of 47 participants. Although a power analysis was performed, this is still a relatively small sample size, which could have had an impact on the results. Future analyses would benefit from the use of a larger cohort. Lastly, both CSO and OO are rich sources of potent bioactive molecules, such as dihydrosterculic acid in CSO and oleocanthal in OO [6,87], which represent additional mechanisms that could influence metabolic health markers. These bioactives were not measured in this study, so we cannot define their impact on the measured results.

## 5. Conclusions

In this study, we applied both untargeted and targeted metabolomics and lipidomics to track changes induced by dietary interventions with CSO and OO in healthy adults. Untargeted metabolomics revealed a large degree of similarity between the CSO and OO participants, suggesting that CSO may produce similar metabolic responses to OO in healthy adults.

Targeted lipidomics revealed that the changes in the acyl chain composition are a direct reflection of the oil consumed. These results suggest that the acyl chain composition may represent a novel biomarker for fat intake. Lipid mediator analysis revealed changes in both pro- and anti-inflammatory LMs in response to high CSO consumption, suggesting that the “one size fits all” assumption about n-6 fatty acids may not be appropriate. Lastly, network analysis revealed subtle drivers of the dietary response and the immune response to the CSO and OO interventions. The primary outcomes of this study were the differential impacts of CSO and OO consumption on the lipid mediator and lipid profiles of healthy adults. The secondary outcomes were the targeted bulk lipidomic analysis, which was not initially planned but undertaken to more deeply explore the lipidome. Further analyses examining the effects of CSO supplementation on human physiology are necessary to pinpoint the mechanisms governing the impact of food intake on human health.

## Figures and Tables

**Figure 1 metabolites-15-00599-f001:**
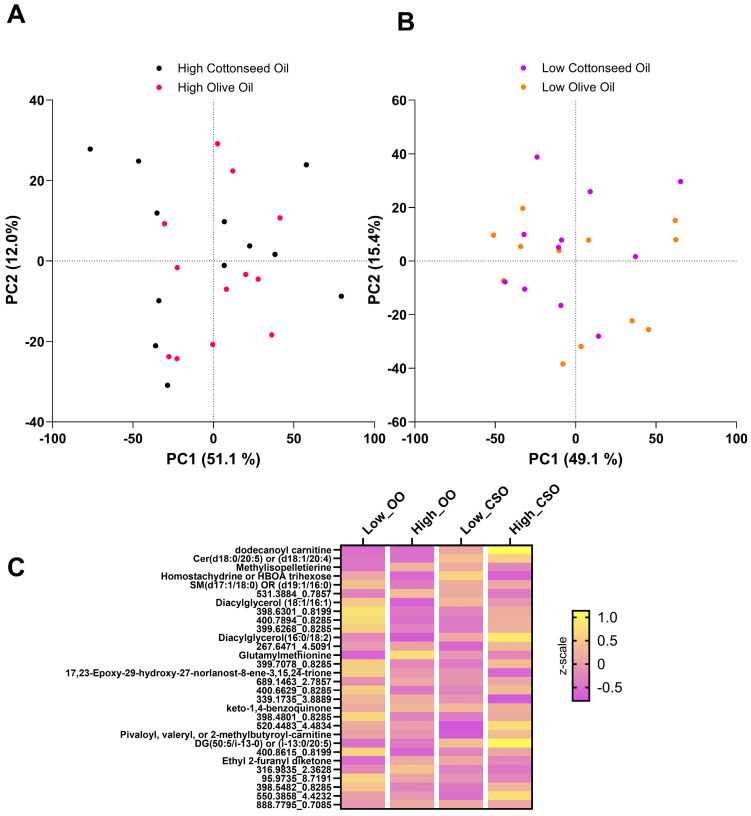
Global multivariate statistics of untargeted metabolomics data. (**A**) Principal component analysis (PCA) score plot of high-dose cottonseed oil (black) and high-dose olive oil (pink) samples. PC1 and PC2 describe 51.1% and 12.0% of the variance in the dataset, respectively. (**B**) PCA score plot of the low-dose cottonseed oil (purple) and low-dose olive oil (orange) samples. PC1 and PC2 explain 49.1% and 15.4% of the variance in the dataset, respectively. (**C**) A heatmap of the metabolomic features that changed significantly (ANOVA, *p* < 0.05) across the intervention groups. Group averages are in the columns, features in rows, and their intersection represents the feature’s abundance relative to the average. Purple indicates low abundance and yellow indicates high abundance. Features that were putatively identified are labeled, and unknown metabolomic features are represented by their m/z value, followed by their retention time in minutes.

**Figure 2 metabolites-15-00599-f002:**
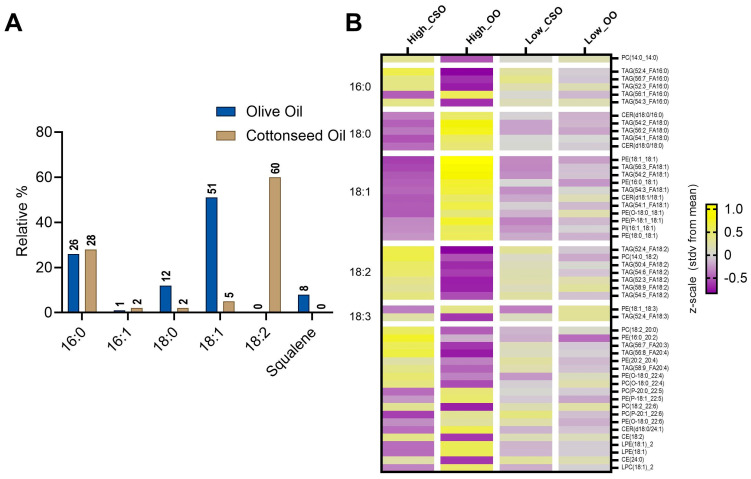
Lipidomic changes reflect the dominant lipid in the dietary oil. (**A**) Fatty acid analysis of the olive oil (blue) and cottonseed oil (tan) used in this study. The dominant fatty acids are represented as a percentage of all fatty acids measured. (**B**) Dietary groups are in columns, lipids are in rows, and their intersection represents the average intensity (z-score) relative to the average across all groups. The displayed lipids have *p* < 0.1 based on ANOVA. Purple indicates low abundance, and yellow indicates high abundance. Lipids are grouped according to their acyl chain length. See Appendix A for a larger version of panel B.

**Figure 3 metabolites-15-00599-f003:**
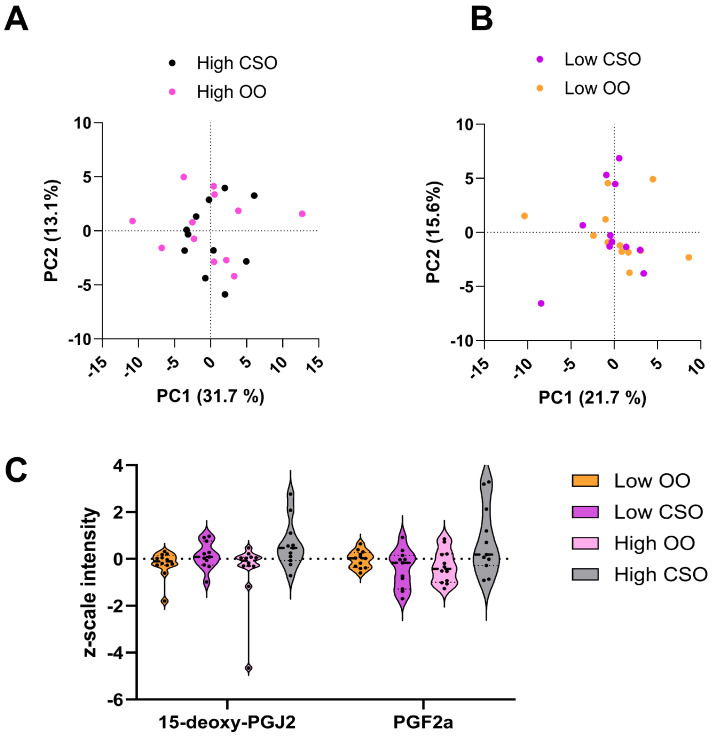
Global multivariate statistics of lipid mediator data. (**A**) Principal component analysis (PCA) score plot of high-cottonseed oil (black) and high-olive oil (pink) samples. PC1 and PC2 explain 31.7% and 13.1% of the variance in the dataset, respectively. (**B**) PCA score plot of the low-cottonseed oil (purple) and low-olive oil (orange) samples. PC1 and PC2 explain 21.7% and 15.6% of the variance in the dataset, respectively. (**C**) Violin plots of the z-scaled abundance of 15-deoxy-PGJ2 and PGF2a, the two lipid mediators that changed significantly (*p* = 0.14, *p* = 0.04) based on ANOVA.

**Figure 4 metabolites-15-00599-f004:**
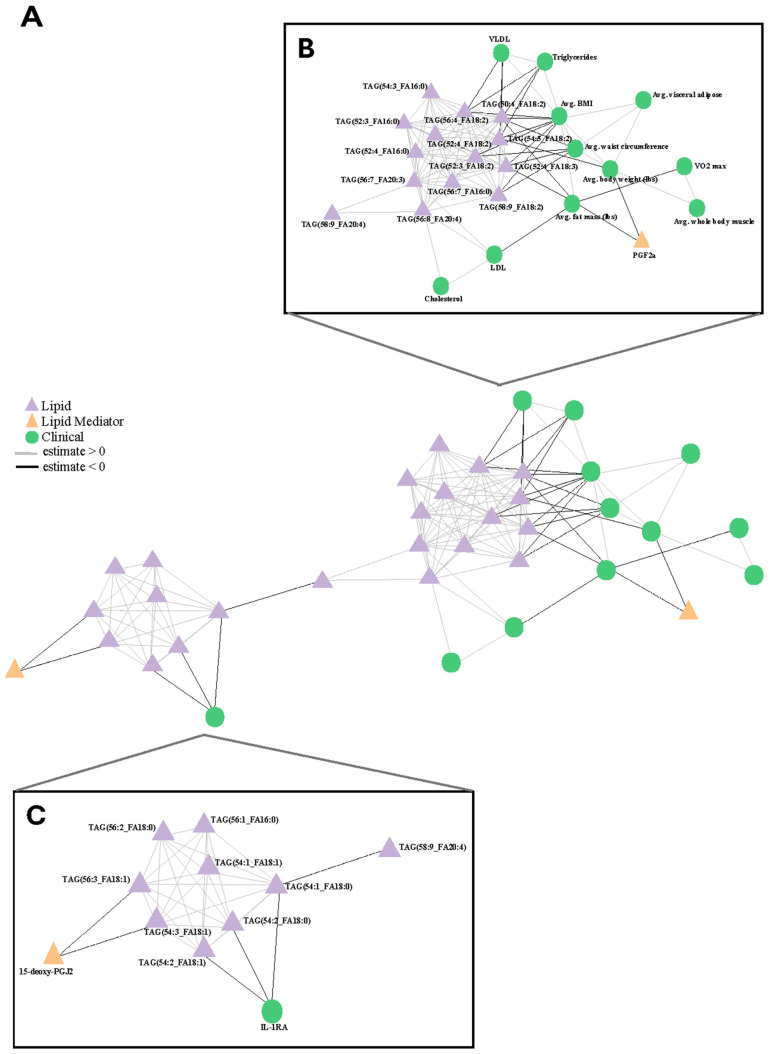
Network analysis of changes induced by CSO and OO interventions. (**A**) Network analysis where purple triangles and orange triangles represent lipids and LMs that changed significantly via ANOVA, respectively; green circles represent clinical variables; black edges represent negative correlations; and gray edges represent positive correlations. (**B**) Inset of cluster 1, including node labels. (**C**) Inset of cluster 2, including node labels. Please see Appendix A for a larger version of this figure.

**Table 1 metabolites-15-00599-t001:** Final participant baseline characteristics. Fasting glucose (mmol/L) and triglyceride (mmol/L) values are presented as the average fasting value from visits 2 and 3. Values are presented as a mean +/− standard deviation (SD).

Participants	Age (years)	BMI (kg/m^2^)	WC (cm)	VAT (L)	Glucose (mmol/L)	Triglycerides (mmol/L)
All (*n* = 47)	31.49 ± 11.83	23.13 ± 1.98	77.40 ± 7.67	0.71 ± 0.57	5.3 ± 0.4	0.93 ± 0.33
Male (*n* = 24)	31.25 ± 12.25	23.58 ± 1.91	80.55 ± 6.86	0.95 ± 0.58	5.4 ± 0.4	1.01 ± 0.4
Female (*n* = 23)	31.74 ± 11.38	22.66 ± 1.94	74.12 ± 7.07	0.46 ± 0.44	5.2 ± 0.3	0.84 ± 0.21

All 47 participants described were included in the metabolomic and lipidomic analyses. Additional participant characteristics can be found in Appendix A.

## Data Availability

The raw data supporting the conclusions of this article will be made available by the authors on request.

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
