# Peer review of "Dietary Intervention with Cottonseed and Olive Oil Differentially Affect the Circulating Lipidome and Immunoregulatory Compounds—A Randomized Clinical Trial"

_metabolites, 2025, doi:10.3390/metabo15090599_

Round 1

Reviewer 1 Report

Comments and Suggestions for Authors

Seed oils are a very relevant topic in the media right now. Therefore, it is important to see studies related to health outcomes, such as cardiometabolic health, on seed oils to ensure that accurate information is being published. This study is interesting, and there is a small, but growing, body of evidence on cottonseed oil and human health outcomes. There are several strengths to the study design and outcome analysis; however, there are also some points that need to be clarified, explained, and/or discussed within the manuscript.

Introduction:

You need to add a hypothesis statement after your purpose statement.

Methods:

I understand why two different flavors of smoothies were used in the study; however, the nutrition content was different between the two flavors. You mention in the methods that they were matched for calories and macros, but that is not correct. Both protein and carbs are different, as well as total calories. Protein is actually doubled for the chocolate vs. mango. That needs to be fixed in the methods when describing the smoothies. Why was this not controlled so the flavors were actual equal in energy, macros, etc.? This needs to be discussed in the paper (perhaps in a limitations paragraph).

Provide more information about the twice daily smoothies vs. 1 smoothie a day. Was the oil equally split between those two smoothies? Why were 2/day used?

The dietary instructions sound like they could have resulted in a significant alteration of daily eating habits of the participants, separate from the smoothies. This is based on all of the foods they had to avoid, as well as now having to cook with either OO or CSO. Did you track how much CSO or OO they used for cooking? If so, how was that factored in to their dose group? If not, this needs to be discussed as the amount of oil used for cooking could have significantly altered how many grams of oil they were consuming each day.

Do you have baseline dietary intake data? What about dietary intake data during the intervention? If so, this data needs to be included. If not, this needs to be added as a discussion point and limitation.

Please clarify what were the primary and secondary outcomes.

What statistics were used for the basic lipid panel and inflammatory markers? I don’t see a stats section, so if you are going to explain each statistical analysis per outcome paragraph, please make sure you don’t skip any of them.

Did you control for physical activity status (either in recruitment or tracking via questionnaire during the study)? Please include this information.

Results:

Do you have the actual cholesterol panel and cytokine data? If so, this should be added to the manuscript.

Discussion:

A limitations paragraph needs to be added (and should include several points brought up in this review, as well as other limitations to the study your group has identified).

Reviewer 2 Report

Comments and Suggestions for Authors

Research Title: Dietary intervention with cottonseed and olive oil differentially affect the circulating lipidome and immunoregulatory compounds

Manuscript ID: metabolites-3783079

The current research article is innovative, interesting and applicable research article in CVD risk management. The authors have very well explained, the overall information about the article is descriptive and informative.  Still some major correction is needed:

However, a few concerns persist.

  1. In Introduction part, line no. 38-43, the author mentioned the Mediterranean Diet but there is very few information about Mediterranean Diet. So, the author should mention the exact composition or principle of Mediterranean Diet.
  2. In Line no. 115-129, Methodology section, sub section “Study Population” The author not clearly mentioned the sample size or total no of volunteer participants who are included in this study.
  3. In Results section, Figure 2. “Lipidomic changes reflect the dominant lipid in the dietary oil” It was observed that there was mentioned Figure (a) 7 (b) but the graph is showing as (A) & (B).
  4. In Discussion Section Line no. 616-617, “Additionally, IL-1RA has been suggested as an early biomarker for progression to type 2 diabetes due to its close association with triglycerides”. The author asked to clarify as IL1RA is an early biomarker for progression to type 2 diabetes.
  5. In references section Line No. 710-711, reference no 15. Please put the journal name.
  6. In references section Line No. 741-742, reference no 27, The journal name is missing. Please incorporate this.
  7. In references section Line No. 751-752, reference no 31, The journal name is also missing. Please incorporate this.

Reviewer 3 Report

Comments and Suggestions for Authors

The manuscript title was “Dietary intervention with cottonseed and olive oil differen tially affect the circulating lipidome and immunoregulatory compounds – a randomized clinical trial”. The research compared the health impact of a diet rich in CSO to olive oil (OO) which is generally considered to be a healthy oil. Specifically, this study examines circulating metabolite and lipid profiles during a 4-week dietary intervention of CSO or OO on 47 healthy adults. The effects of CSO and OO on immunoregulatory was different about the metabolites. The research was meaningful. The relationship between the the lipids, LMs, and clinical covariates were also analyzed and discussed. My advice was that the discussion can be separated and inserted in the results.

Author Response

REVIEWER 3

Comments and Suggestions for Authors

The manuscript title was “Dietary intervention with cottonseed and olive oil differen tially affect the circulating lipidome and immunoregulatory compounds – a randomized clinical trial”. The research compared the health impact of a diet rich in CSO to olive oil (OO) which is generally considered to be a healthy oil. Specifically, this study examines circulating metabolite and lipid profiles during a 4-week dietary intervention of CSO or OO on 47 healthy adults. The effects of CSO and OO on immunoregulatory was different about the metabolites. The research was meaningful. The relationship between the the lipids, LMs, and clinical covariates were also analyzed and discussed. My advice was that the discussion can be separated and inserted in the results.

Thank you for your positive feedback. We agree that combining the discussion and results could help with the flow of the manuscript; however, it is a journal requirement to keep them separate.

Reviewer 4 Report

Comments and Suggestions for Authors

Cooper et. al. applied LC-MS method to focus on the lipidome change in plasma which revealed the lipidome change between adult intaking CSO and OO. I do have couple comments on the current manuscript.

  1. Abstract, Line 11, Linoleic acid (18:2) should be written as FA 18:2, since this manuscript mainly talk about lipidomics, I would suggest all lipid nomenclature should follow Lipid Standard Initiative (DOI: 10.1194/jlr.S120001025). Check the entire manuscript.
  2. I did not see any information about the MS data normalization. How did the author make that? And by using what? The internal standard which did not be mentioned or pooled sample?
  3. A statistics section should be added. The author should introduce what algorithms used. One way ANOVA followed by what post hoc test to differentiate metabolites between different groups, etc.

Reviewer 5 Report

Comments and Suggestions for Authors

General comments: 

It is well written and uses appropriate methods. It describes a clinical trial approved by a bioethics committee with a relevant number of subjects (47). The manuscript reaches conclusions that somewhat contradict the prevailing consensus on the effects of saturated/monounsaturated fat in the human diet, although there are some factors that must be taken into account for these conclusions, such as the inclusion of two beverages containing fiber and other bioactive components of plant origin in a population that did not previously consume them. The proportion of olive or coconut oil in the participants' daily fat intake is also somewhat debatable. Nevertheless, the manuscript is well written and the results deserve to be published, although personally I do not entirely agree that the results obtained are entirely reproducible.

Specific comments:

Line 11: “PUFA” must be defined. The same for PGF2a in line 24. 

Line 28-40. The attribution of health properties to olive oil in only in part caused by olecic acid, but also for minor compounds, such as oleocanthal, oleuropein, etc. 

Line 47: monounsaturated was previously cited in line 40. MUFA should be defined in line 40 rather than 47. 

Line 64: Lowering the ratio n-6/n-3 to a 4-5:1 ratio, not all decrease in beneficial. 

Lines 90-105. Please check spacing arround braquets. 

Line 117: Please define “MT”. The same in line 128 for AHA and ACSM. Please, check all the manuscript acromyns usage. 

Line 172: Please use grams instead of “oz”. 

Line 211 was cited “minute” and in line 212 “min”. Both forms are possible, but please, be consistent. The same in lines 229-233. 

Lines 309-311. This is materials and methods, not a results. 

Figure 2 can not be read properly. Its needs to make bigger the scale text. The same for Figure 4B. The text cannot be read properly. 

Lines 451-454: TAG, LDL, VLDL...although it is obvious, the significance of abbreviations must be stated the forts time that appears in the text.  

A limitations of the study section in always wellcome and should be included. The number of subjects in under 50, and the inclusion of two daily smooties can be a substantial modification of the diet itshelf (among others). 

There are a lot of mistakes in references. In example, references 2,15,16 20,22,25,27...etc lacks journal name. 

Round 2

Reviewer 1 Report

Comments and Suggestions for Authors

The authors have addressed my comments.

Author Response

Thank you for your helpful suggestions. All changes have been made. 

Reviewer 2 Report

Comments and Suggestions for Authors

Comments and Suggestions for Authors

Research Title: Dietary intervention with cottonseed and olive oil differentially affect the circulating lipidome and immunoregulatory  compounds – a randomized clinical trial

Manuscript ID: metabolites-3783079

The current research article is innovative, interesting and applicable research article in lipidome changes by addition of cotton seed oil and olive oil differently. The authors have very well explained, the overall information about the article is descriptive and informative.  Still some major correction is needed:

However, a few concerns persist.

  1. In Introduction part, line no. 38-43, the author justified the statement of the Mediterranean Diet but there is very few information about Mediterranean Diet.
  2. In Line no. 121-138, Methodology section, sub section “Study Population” The author now revised & mentioned the sample size or total no of volunteer participants who are included in this study.
  3. In Table no. 1 Line no 341-344, the author should clarify the unit of Blood glucose and triglyceride level in mmole/L in spite of mg/dl.
  4. In Discussion Section Line no. 616-617, “Additionally, IL-1RA has been suggested as an early biomarker for progression to type 2 diabetes due to its close association with triglycerides”. The author already mentioned the clarification as IL1RA is an early biomarker for progression to type 2 diabetes and incorporated the reference.
  5. In references section Line No. 759-761, reference no 14. Please put the DOI number.
  6. In references section Line No. 765-766, reference no 16, The DOI is missing. Please incorporate this.
  7. In references section Line No. 767-768, reference no 17, The journal name is also missing. Please incorporate this.
  8. In references section Line No. 837, reference no 43, The reference is very old and doi number is also missing.

Author Response

  1. In Introduction part, line no. 38-43, the author justified the statement of the Mediterranean Diet but there is very few information about Mediterranean Diet.

Thank you for this suggestion. We have incorporated more information about the Mediterranean diet. This passage now reads:

Lines 38-47:

The Mediterranean diet, which emphasizes plant-based foods and up to eight servings of olive oil daily, has also been recognized as a dietary intervention capable of reducing CVD [3,4]. The Mediterranean diet does not limit calories or cut entire food groups out, but instead emphasizes consuming an abundance of plant-based foods and fresh fruit daily, olive oil as the primary fat source, limits dairy and red meat consumption, and suggests moderate amounts of fish and poultry [3]. The beneficial health effects of a Mediterranean diet are often contributed to olive oil (OO), which is high in the monounsaturated (MUFA) fat, oleic acid (FA 18:1) [4] in addition to other bioactive antioxidants and polyphenols such as oleocanthal, oleuropein, and tocopherols [5-7].

  1. In Line no. 121-138, Methodology section, sub section “Study Population” The  author now revised & mentioned the sample size or total no of volunteer participants who are included in this study.

Thank you for recognizing our change to address previous comments.

  1. In Table no. 1 Line no 341-344, the author should clarify the unit of Blood glucose and triglyceride level in mmole/L in spite of mg/dl.

We have added the following to clarify that the units of blood glucose and triglycerides are in mmol/L:

Lines 343-344: “Fasting glucose (mmol/L) and triglyceride (mmol/L) values are presented as the average fasting value from visits 2 and 3.:

  1. In Discussion Section Line no. 616-617, “Additionally, IL-1RA has been suggested as an early biomarker for progression to type 2 diabetes due to its close association with triglycerides”. The author already mentioned the clarification as IL1RA is an early biomarker for progression to type 2 diabetes and incorporated the reference.

Thank you for this comment.

  1. In references section Line No. 759-761, reference no 14. Please put the DOI number.

The doi has been added to the following reference:

Line 760:

Patterson, E.; Wall, R.; Fitzgerald, G.F.; Ross, R.P.; Stanton, C. Health implications of high dietary omega-6 polyunsaturated Fatty acids. 2012, 2012, 539426, doi: https://doi.org/10.1155/2012/539426

  1. In references section Line No. 765-766, reference no 16, The DOI is missing. Please incorporate this.

The doi has been added to the following reference:

Line 767-768:

Corsinovi, L.; Biasi, F.; Poli, G.; Leonarduzzi, G.; Isaia, G. Dietary lipids and their oxidized products in Alzheimer's disease. Molecular Nutrition & Food Research 2011, 55, doi: https://doi.org/10.1002/mnfr.201100208

  1. In references section Line No. 767-768, reference no 17, The journal name is also missing. Please incorporate this.

The journal name has been added to the following reference:

Line 771-772:

Innes, J.K.; Calder, P.C. Omega-6 fatty acids and inflammation. Prostaglandins, Leukotrienes, and Essential Fatty Acids 2018, 132, 41-48, doi:https://doi.org/10.1016/j.plefa.2018.03.004.

  1. In references section Line No. 837, reference no 43, The reference is very old and doi number is also missing.

The reference has been updated.

Line 841-842:

Csárdi, G., & Nepusz, T. The Igraph Software Package for Complex Network Research. InterJournal, Complex Systems 2006, 1695. doi: https://doi.org/10.5281/zenodo.3630268